# A Box–Behnken Extraction Design and Hepatoprotective Effect of Isolated Eupalitin-3-*O-β-D*-Galactopyranoside from *Boerhavia diffusa* Linn.

**DOI:** 10.3390/molecules27196444

**Published:** 2022-09-29

**Authors:** Kamal Y. Thajudeen, Yahya I. Asiri, Shahana Salam, Shabeer Ali Thorakkattil, Mohamed Rahamathulla, Ilyas Uoorakkottil

**Affiliations:** 1Department of Pharmacognosy, College of Pharmacy, King Khalid University, Abha 61441, Saudi Arabia; 2Department of Pharmacology, College of Pharmacy, King Khalid University, Abha 61441, Saudi Arabia; 3Department of Pharmaceutical Chemistry, College of Pharmacy, Prince Sattam bin Abdulaziz University, P.O. Box 173, Al-Kharj 11942, Saudi Arabia; 4Pharmacy Services Department, Johns Hopkins Aramco Healthcare, Dhahran 31311, Saudi Arabia; 5Department of Pharmaceutics, College of Pharmacy, King Khalid University Greiger, Abha 61421, Saudi Arabia; 6Department of Pharmacognosy and Phytochemistry, Moulana College of Pharmacy, Malappuram 679321, Kerala, India

**Keywords:** *Boerhaviadiffusa* Linn., eupalitin-3-*O-β-D*-galactopyranoside, optimization, HPTLC, hepatoprotective activity

## Abstract

The objectives of this study were to optimize and quantify the maximum percentage yield of eupalitin-3-*O-β-D*-galactopyranosidefrom *Boerhavia diffusa* leaves using response surface methodology (RSM), as well as to demonstrate the hepatoprotective benefits of the bioactive compound. The Box–Behnken experimental design was utilized to optimize the eupalitin-3-*O*-*β*-*D*-galactopyranoside extraction procedure, which also looked at the extraction duration, temperature, and solvent concentration as independent variables. *Boerhaviadiffusa* leaves were extracted, and *n*-hexane, chloroform, ethyl acetate, and water were used to fractionate the dried extracts. The dried ethyl acetate fraction was thoroughly mixed in hot methanol and stored overnight in the refrigerator. The cold methanol was filtered, the solid was separated, and hot methanol was used many times to re-crystallize the solid to obtain pure eupalitin-3-*O-β-D*-galactopyranoside (0.1578% *w*/*w*). The proposed HPTLC method for the validation and quantification of eupalitin-3-*O-β-D*-galactopyranosidewassuccessfully validated and developed. The linearity (*R_2_* = 0.994), detection limit (30 ng), and quantification limit (100 ng) of the method, as well as its range (100–5000 ng), inter and intraday precision (0.67% and 0.991% RSD), specificity, and accuracy (99.78% RSD), were all validated as satisfactory. The separation of the eupalitin-3-*O-β-D*-galactopyranoside band was achieved on an HPTLC plate using toluene:acetone:water (5:15:1 *v*/*v*) as a developing system. The Box–Behnken statistical design was used to determine the best optimization method, which was found to be extraction time (90 min), temperature (45 °C), and solvent ratio (80% methanol in water *v*/*v*) for eupalitin-3-*O-β-D*-galactopyranoside. Standard silymarin ranged from 80.2% at 100 µg/mL to 86.94% at 500 µg/mL in terms of significant high hepatoprotection (cell induced with carbon tetrachloride 0.1%), whereas isolated eupalitin-*3-O-β-D*-galactopyranoside ranged from 62.62% at 500 µg/mL to 70.23% at 1000 µg/mL. More recently, it is a source of structurally unique flavonoid compounds that may offer opportunities for developing novel semi-synthetic molecules.

## 1. Introduction

*Boerhaviadiffusa* (BD) Linn., also known as Punarnava (Syn. *Boerhavia repens* Linn., *Boerhavia procumbens* Roxb., Family Nyctaginaceae), is an important rejuvenatingdrug used in Ayurveda. It is widely distributed in many tropical and subtropical countries, including India, Sri Lanka, Egypt, Sudan, Ghana, Nigeria, China, Australia, the Philippines, and Iran. It is a diffusely branched, low spreading perennial herb or under-shrub with an elongate fusiform tuberous tap root and branches often reddish in color. Its leaves are simple, ovate, or oblong–obtuse to nearly sub orbicular, in unequal pairs, 2–4 × 2–4.5 cm, thick, green glossy above, silvery white beneath, with small pink flowers on a long axillary peduncled umbellate cluster or in terminal panicles. It takes on a perianth funnel form, constricted in the middle with 2–3 stamens. This plant’s roots and leaves, in particular, have been widely used in folk medicine to treat various illnesses, particularly those affecting the gastrointestinal tract. Specifically, this plant has been reported for the inhibition of human cervical cancer [1], anti-hyperglycaemic and reno-protective effects [2], antidepressant activity, anticonvulsant and antiepileptic characteristics [3,4], chemo-preventive action [5], inhibitory effect of prostatic hyperplasia and potent anti-breast cancer activity [6,7], genotoxic and antigenotoxic activity [8], antiproliferative and antiestrogenic effects [9], thrombolytic, cytotoxic, and anti-microbial activities [10], immunomodulatory [11] and anti-metastatic activity [12], anti-diabetic activity [13], anti-urolithic activity [14], Ca^2+^ channel antagonistic activity [15], arsenic trioxide-induced cardiotoxicity [16] and anti-inflammatory activity [17]. Hepatitis, hepatic necrosis, and jaundice are major liver ailments with an elevated mortality toll. Currently, only rare hepatoprotective allopathic medicines are accessible for the management of liver ailments. Therefore, people are using different extracts from plants for the management of liver disorders. Many plant-based drugs, such as *Andrographis paniculata*, *Boerrhaviadiffusa*, *Calotropis procera*, *Fumaria indica*, *Garcinia cambogia*, *Luffa acutangula*, *Mamordicasubangulata*, *Naragamiaalata*, *Nigella sativa*, and *Trigonella foenum graecum*, have been found to be hepatoprotective in traditional medicine [18]. Secondary metabolites in BD include flavonoid glycosides, isoflavonoids (rotenoids), steroids (ecdysteroids), alkaloids, and phenolic and lignan glycosides. *Boerhavia diffusa* contains a high concentration of flavonoid glycosides, which may have antihypertensive and hepatoprotective properties. Several flavonoid glycosides are isolated from the leaves of *Boerhavia diffusa* that show potent anticancer [19], immunomodulatory [20], atherosclerosis [21], antihypertensive activity [22,23], spasmolytic activity [24], and antioxidant properties by scavenging free radicals and increasing glutathione concentrations, allowing them to be used in the treatment of hepatitis and hepatic cirrhosis [25]. The yields of the yellow crystalline powder of eupalitin-3-*O*-*β*-*D*-galactopyranoside indicate that *Boerhavia diffusa* leaves are a rich source of flavonoid glycoside moiety. Only one natural product, silymarin, has been used for hepatoprotection to date, but silymarin is a flavonoid glycoside that is structurally similar to eupalitin-3-*O-β-**D*-galactopyranoside. We chose eupalitin-3-*O-β-**D*-galactopyranoside because of its potent hepatoprotective activity against carbon tetrachloride-induced toxicity. The compound is purified by direct crystallization from the residue obtained after the sedimentation method, making it the simplest, quickest, mosteconomically viable, scale-up compatible for commercial utilization, and novel method for the isolation of compounds of eupalitin-3-*O*-*β*-*D*-galactopyranoside with a yield of up to 0.1578% *w*/*w*, which is more than 100 times the yield reported by Li et al. [26]. The chemical structure of eupalitin-3-*O-β-**D*-galactopyranoside is shown in Figure 1.

Several analytical methods, including high-performance liquid chromatography (HPLC), high-performance thin layer chromatography (HPTLC), liquid chromatography-mass spectrometry (LCMS), high-performance liquid chromatography (HPLC), and ultra-performance liquid chromatography (UPLC), can be used to determineflavonoid glycosides in medicinal plants. Eupalitin-3-*O*-*β*-*D*-galactopyranoside (flavonoid glycoside) is rarely applied by the use of similar analytical techniques. In 2005, Ferreres et al. reported the identification and quantification of flavonoid glycosides using HPLC–PAD-ESI/MS [27]. A validated high-performance thin layer chromatography (HPTLC) method for validation, quantification, and optimization of eupalitin-3-*O*-*β*-*D*-galactopyranoside from the hydro-alcoholic extract of *B. diffusa* was developed using a single solvent system (toluene: acetone: water (5:15:1). Because of advantages such asthe ease of sample preparation, optimization of specific chemicals, and comparison of several samples on a single plate, comparable chromatographic techniques are frequently used to evaluate retinoid and phenolic acids. Comparing entire chromatograms allows for the detection of minor similarities and differences between the plants under investigation. For in vitro models of normal liver cells, human hepatoma cell lines are proposed as an alternative to human hepatocytes. HepG2 hepatoma cell lines are widely used in research on liver function, metabolism, and drug toxicity [28,29]. HepG2 cells have many of the same biochemical and morphological properties as normal hepatocytes [30]. These cells are used in studies to determine whether medicinal plants have hepatoprotective properties because they retain many characteristics of normal liver cells [31,32]. Gonzalez LT et al. reported the hepatotoxic effects of cyclosporine A on HepG2 cells in 2017, focusing on changes in the activation pattern of antioxidant enzymes glutathione peroxidase (GPx) and glutathione reductase (GR), glutathione (GSH) levels, intracellular ROS production, supernatant alanine aminotransferase (ALT), and aspartate aminotransferase (AST) [33]. The objective of the study was to optimize, validate, and quantify eupalitin-3-*O-β-D*-galactopyranosidein hydro-alcoholic extracts of *B. diffusa* and also investigate the hepatoprotective activity of isolated eupalitin-3-*O-β-**D*- galactopyranoside.

## 2. Results

### 2.1. Extract and Percentage Yield of Eupalitin-3-O-β-D-Galactopyranoside

The crude hydro-alcoholic extract of *B. diffusa* yielded 9.9% *w*/*w* and was subsequently fractionated using different organic solvents based on polarity index differences. Non-polar *n*-hexane yielded 33% of the fraction, while chloroform and medium polar ethyl acetate yielded 17% and 11%, respectively. The solid yellow crystalline powder of eupalitin-3-*O-β**-D*-galactopyranoside yielded 0.1578% *w*/*w*.

### 2.2. Development of the Mobile Phase

For chromatographic separation studies, a standard working solution of eupalitin-3-*O*-*β*-*D*-galactopyranoside (1 mg/mL) in methanol was used. Initially, multiple solvent solutions were used in various trials. Finally, toluene: acetone: water (5:15:1) for determination of eupalitin-3-*O-β-**D*-galactopyranosideprovideda well-defined and sharp peak. After 30 min of saturating the chamber with the mobile phase at room temperature, we obtained well-defined bands. The standard band of eupalitin-3-*O-β-**D*-galactopyranoside and hydro-alcoholic extracts of *B. diffusa* was presented on an HPTLC plate scanned at 340 nm (Figure 2).

### 2.3. Method of Validation

Figure 2 shows a representative chromatogram of eupalitin-3-*O-β**-D-* galactopyranoside in the established HPTLC technique. The HPTLC chromatogram in Figure 3 shows a retention factor of eupalitin-3-*O*-*β*-*D*-galactopyranoside (R_f_ = 0.56).

The calibration curve’s linear regression data revealed an excellent linear relationship throughout a concentration range of 100 to 5000 ng mL^−1^ with a correlation coefficient (*R*^2^) value of 0.9984, indicating acceptable linearity (Table 1). A high correlation coefficient value (a value extremely close to 1.0) indicates a strong linear relationship between the peak area and concentration of eupalitin-3-*O-β**-D*-galactopyranoside. The LOD and LOQ were calculated using the ICH Guidelines Q2 (R1) (2005) and were 30 ng and 100 ng for eupalitin-3-*O-β**-D*-galactopyranoside (Table 1). The intra-day and inter-day assays were used to determine the repeatability of the proposed HPTLC method, and the results were expressed in terms of % RSD (Table 1). The low-percent RSD suggested the developed HPTLC method’s outstanding precision for repeatability and intermediate precision. The specificity of the developed HPTLC method for the analysis of eupalitin-3-*O*-*β*-*D*-galactopyranoside in the hydro-alcoholic extract was confirmed by comparing the spectra obtained in the standards and sample extract analyses. This spectra’s peak, apex, peak start, and peak end positions were identical. The recovery trials were carried out to determinehow sensitive the method was for estimating eupalitin-3-*O-β**-D*-galactopyranoside. The standard addition approach was used to increase the concentration of eupalitin-3-*O-β-**D*-galactopyranoside in the sample extract by 50%, 100%, and 150%. The percentage recoveries for eupalitin-3-*O*-*β*-*D*-galactopyranoside ranged from 98.88% to 100.68% for the three concentrations, indicating a high level of precision. Table 1 shows the percentage average recovery value. The analytical method was highly accurate, as evidenced by the near-100 percent mean percent recovery and low percent RSD values.

### 2.4. Box–Behnken Design-Experiment 

A three-level factorial, Box–Behnken applied statistical experimental approach was performed using 17 experimental trial runs. The quadratic model was determined to be the best-fitting model for standard eupalitin-3-*O-β-**D*-galactopyranoside, and the comparative values of R, SD, and percent C.V for the various planned models are presented in Table 2, along with the regression equations usually utilized for eventually elite replies.

Only statistically significant (*p* < 0.0001) coefficients for eupalitin 3-*O-β-**D*-galactopyranoside are involved in the equations. A positive effect implies that the effect favors the optimization of eupalitin-3-*O-β-**D*-galactopyranoside; a negative value indicates that the factor and the response are inversely related. The equations show that extraction time (A_1_), extraction temperature (B_2_), and solvent ratio (water: alcohol) (C_3_) have a negative effect on the reaction; while extraction time (A_1_), extraction temperature (B_2_), and solvent ratio (C_3_) have a positive effect on eupalitin-3-*O-β-**D*-galactopyranoside (Y_1_). The equations show that the factors time (B_1_) and temperature (B_2_) have an unresponsive effect, and 40:60:80 “*v*/*v/v”* solvent ratios (B_3_) have a productive effect on the responses (A_1_ and B_2_). The equations show that the factors time (A_1_) and temperature (B_2_) have an uncooperative effect, and 40:60:80 *v*/*v* solvent ratios (C_3_) have a productive effect on the responses (A_1_ and B_2_). It also indicates that the correlation between factors and response is not always linear. When more than one factor is replaced at the same time, a factor can represent various degrees of response. Relationships between A_1_ and B_2_, as well as between A_1_ and C_3_, induce uncooperative impacts on the response. However, the result of the square root of various factors does not repeat history, as shown by its performance. In the case of the square root of various factors, B_2_^2^ (extraction temperature) and C_2_^3^ (solvent concentration) produced positive results, while A_2_^1^ (time of extraction) produced negative results. The last mixture ratios of the extractions were established based on the percentage yield of eupalitin-3-*O*-*β*-*D*-galactopyranoside. Using three D-response surface plots (Figure 4), the final compositional ratios of the extractions were chosen based on the percentage yield of eupalitin-3-*O-β-**D*-galactopyranoside (Figure 4).

### 2.5. Effect of Independent Factors on Eupalitin-3-O-β-D-Galactopyranoside (Y_1_)

Regression equation of the fitted models:Y_1_ = 0.035−0.00097 A_1_+ 0.006 B_2_ + 0.0031 C_3_ + 0.0038 A_1_ B_2_ + 0.0043 A_1_ C_3_ − 0.0051 B_2_ C_3_ …(A_1_)

The extraction time (Factor A_1_) affected the yield of eupalitin-3-*O*-*β*-*D*-galactopyranoside. It was observed that equation A_1_ had a highly positive effect on eupalitin-3-*O-β-**D*-galactopyranoside as compared to B_2_ and C_3_. With the increased time of extraction, the yield of eupalitin-3-*O-β-**D*-galactopyranoside also increased due to the time required to penetratethe solvent into the plant materials. We increased the yield of eupalitin 3-*O*-*β*-*D*-galactopyranoside by extending the extraction time. However, it dropped after reaching a peak, which may be due to the compound becoming saturated, as shown in Figure 4A.

Factor B_2_ (extraction temperature) had a greater positive effect on eupalitin-3-*O-β-**D*-galactopyranosideyield than factors A_1_ and C_3_. As the temperature increased, the yield of eupalitin-3-*O-β-**D*-galactopyranosidealso increased due to the thermo-stable compound, as shown in Figure 4B. After the temperature reached its peak, degradation caused the percentage yield of eupalitin-3-*O-β-**D*-galactopyranosideto drop.

Factor C_3_ (solvent ratio % (alcohol: water) showed a positive effect on eupalitin-3-*O*-*β*-*D*-galactopyranoside percentage yield compared to factor B_2_. It was observed that the solvent ratio was higher, and the yield of eupalitin-3-*O*-*β*-*D*-galactopyranoside was higher. The medium polarity of the compound may also be responsible for this, as shown in Figure 4C. The hydro-alcoholic extract (80%) was found to have a higher concentration of eupalitin-3-*O-β-**D*-galactopyranoside than other extracts. This could be because hydro-alcoholic extract contains medium-polar compounds. Temperature, time, solvents of extraction, and the chemical moiety of secondary metabolites all influence the percentage yield of extracts. The solvent utilized and the chemical properties of the sample are the most relevant factors under the same time and temperature conditions. The use of a Box–Behnken expert experiment to optimize eupalitin-3-*O-β-**D*-galactopyranoside in a single mobile solvent based on temperature (°C), duration (min), and solvent ratio (*v*/*v*) had a direct influence on eupalitin-3-*O*-*β*-*D*-galactopyranoside.

### 2.6. Identification of Eupalitin-3-O-β-D-Galactopyranoside

The UV spectrum (340 nm) of eupalitin-3-*O*-*β*-*D*-galactopyranoside revealed flavonoid moiety bands. When sodium methoxide was added, the band underwent a bathochromic shift (+45 nm), revealing a hydroxyl group in position 3. The EIMS of compound A revealed a molecular ion peak at *m*/*z* 330 (M) + and two prominent ion peaks at *m*/*z* 121 (B+) and 184 (A+), indicating the presence of one hydroxyl, two methoxyl, and one hydroxyl group in the B-ring and one hydroxyl group in the A-ring, respectively. 1H-NMR (DMSO-d_6_) δ:12.560 (1H, 5-OH), 8.111 (2H, H-2′ and H-6′), 6.880 (2H, H-3′ and H-5′), 6.601 (1H, H-8), 4.751 (3H, 6-OCH_3_), 4.3930 (3H, 7-OCH_3,_ 5.411 (1H, H-1″), and 4.141–6.701 (sugar protons). 1H-NMRsignals at 8.11 (2H, d, *J* = 9HZ) and 6.88 (2H, d, *J* = 9HZ) strongly suggested the presence of four hydroxyls with no other substitution in the B ring and also exposed the presence of two methoxyl groups (4.751 (3H, 6-OCH3), 4.3930 (3H, 7-OCH3).A hydroxyl group at fivepositions (12.560 (1H, 5-OH)) and an anomeric sugar proton 5.411 (1H, H-1″), d, *J* = 7.5HZ) were also determined. ^13^C-NMR (60 MHZ, Pyridine-D5)δ: 57.3 (7-OCH3), 60.9 (6-OCH3), 92 (C-8), 159.3 (C7), 132.7 (C-6), 153.0 (C-5), 179.0 (C-4), 135 (C-3), 157.7 (C-2), 104.2 (C-1″), 73.7 (C-2″), 75.2 (C-3″), 69.7 (C-4″), 77.5 (C-5″), 61.8 (C-6″), 121.9 (C-1′), 131.9 (C-2′), 116.1 (C-3′), 161.7 (C-4′), 116.1 (C-5′), and 131.9 (C-6′). According to the information and literature, the isolated chemical was identified as eupalitin-3-*O-β-**D*-galactopyranoside [34,35].

### 2.7. In Vitro Cytotoxicity Study of Eupalitin-3-O-β-D-Galactopyranoside

The drug sample was tested forin vitro cytotoxicity against the HepG2 cell line. To determine the percentage growth inhibition of the drug on cell lines, drugs were taken at concentrations ranging from 100 to 1000 µg/mL. The drug sample had a CTC_50_ (concentration required to reduce viability by 50%) value greater than 1000 µg/mL.

### 2.8. Hepatoprotective Activity of Eupalitin-3-O-β-D-Galactopyranoside

In the study, the above-mentioned fractions were screened for hepatoprotective activity against CCl_4_-induced cytotoxicity by pre-incubating the cells with or without the extracts or silymarin. A significant decrease in cell viability was observed upon treatment of HepG2 cells with CCl_4_ (0.1%). Results are expressed as mean ± standard error mean (S.E.M). A sample concentration providing percentage hepatoprotection of standard silymarin (84 percent at 500 µg/mL), purified eupalitin-3-*O*-*β*-*D*-galactopyranoside (70.23 percent at 1000 µg/mL), chloroform fraction (23 percent at 1000 µg/mL), and remaining ethyl acetate fraction (42 percent at 500 µg/mL) was calculated from graph plotting. In Figure 5 and Table 3, the % protection is graphically represented.

## 3. Discussion 

The literature review demonstrates that many natural products with hepatoprotective properties require standardization and optimization because polyphenolic components degrade during the extraction process. Compared to alcoholic and aqueous extracts, the hydro-alcoholic extract was shown to have a relatively higher concentration of eupalitin-3-*O-β-D*-galactopyranoside. This might be due to the hydro-alcoholic extract’s presence of medium-polar compounds (flavonoid glycosides). The temperature, duration, and solvent ratio (alcohol: water) used during extraction and the chemical moiety of secondary metabolites all affect the percentage yield of eupalitin-3-*O-β-D*-galactopyranoside. The Box–Behnken design is used in the current study to optimize the extraction of eupalitin 3-*O*-*β*-*D*-galactopyranoside based on temperature (°C), time (min), and solvent concentration ratio (percent *v*/*v*). In patent EP 1485394 A1, procedures for isolating eupalitin-3-*O-β-D*-galactopyranosideare isolated using preparative TLC. In our current study, *Boerhavia diffusa* leaves were extracted, and the dried extracts were then fractionated using *n*-hexane, chloroform, ethyl acetate, and water. The dried ethyl acetate fraction was thoroughly mixed in hot methanol and stored overnight in the refrigerator. The cold methanol was filtered, the solid was separated, and hot methanol was used many times to re-crystallize the solid to obtain pure eupalitin-3-*O-β-D*-galactopyranoside. This isolated compound was used as a marker compound for the determination and quantification of eupalitin-3-*O-β-D*-galactopyranoside. Only the plant extract (*Boerhavia diffusa*) was found to have hepatoprotective properties, and no one has yet documented isolated eupalitin-3-*O*-*β*-*D*-galactopyranoside to have such effects. Yellow amorphous solids were characterized as eupalitin-3-*O-β-D*-galactopyranoside using spectroscopic studies and comparison with published data. It is also characterized as eupalitin-3-*O*-*β*-*D*-galactopyranoside when matched with an authentic sample of eupalitin-3-*O*-*β*-*D*-galactopyranoside by HPTLC. Only HPTLC methods for determining eupalitin-3-*O-β-D*-galactopyranoside have been reported [36]. None of them have demonstrated their specificity and optimization of eupalitin-3-*O*-*β*-*D*-galactopyranoside in *B. diffusa*. The proposed method is innovative since it is the first analytical method to report on the optimization of eupalitin-3-*O-β-D*-galactopyranoside in a hydro alcohol extract of *Boerhavia diffusa* Linn using the Box–Behnken statistical design. Nowadays, optimization of eupalitin-3-*O-β-D*-galactopyranoside is very important for the isolation of eupalitin-3-*O-β-D*-galactopyranoside from plant materials due to the degradation of flavonoid glycoside. The yields of the yellow crystalline powder of eupalitin-3-*O-β-D*-galactopyranosideindicated that the *Boerhavia diffusa* leaves are a rich source of flavonoid glycoside moiety. However, the bioactive elements that are responsible for their therapeutic effectiveness are unknown. In India, medicinal herbs are a key source of hepatoprotective drugs. More than 700 mono and polyherbal preparations in the form of decoctions, tinctures, and pills are used for various liver ailments [37]. Ayurveda is the earliest Indian system of medicine, defining many plants for managing hepatitis and hepatotoxicity. Traditionally, plants have been used therapeutically for a variety of conditions [38]. Compared to the narrow spectrum effect of synthetic drugs with the associated danger of side effects; traditional drugs show minor action and require long-term administration to be effective, mainly in chronic conditions [39]. *Boerhavia diffusa* extract has been proven to be hepatoprotective in nature [40]. Currently; only rare hepatoprotective allopathic medicines are accessible for managing liver ailments. As a result, people are adapting various plant extracts to treat liver syndromes. However, it is the flavonoids and flavonoid glycosides that are responsible for hepatoprotection. There is now just one protective natural medicine (silymarin), which is neither curative nor without limits in terms of protecting the liver from viral infections. In 2011, Karimi G. et al. found silymarin to be effective in treating liver cirrhosis and hepatitis, as well as having immunomodulatory properties [41]. However, the flavonoid moiety was present in the silymarin molecule. Free radicals are captured by flavonoids, which reduce lipid peroxidation [42]. The presence of flavonoids in *Boerhavia diffusa* is frequently linked to its biological action. Flavonoid glycosides come in various forms, each with their own health benefits [43]. As a result, isolated eupalitin-3-*O-β-D*-galactopyranoside demonstrated a considerable hepatoprotective effect in HepG2 cells produced with carbon tetrachloride toxicity compared to standard silymarin. In the present study, the validated high-performance thin layer chromatography (HPTLC) method was a newly developed solvent system for validating, quantifying, and optimizing eupalitin-3-*O-β-D*-galactopyranoside in *B. diffusa*. The HPTLC method chosen is accurate, exact, easy, specific, time-saving, and cost-effective, and it can distinguish eupalitin-3-*O-β-D*-galactopyranoside molecules from other ingredients.

## 4. Materials and Methods

### 4.1. Chemicals and Equipment

Eupalitin-3-*O-β-D*-galactopyranosidewas obtained from Hamdard laboratory, New Delhi, India. To prepare standards and samples, methanol of HPLC grade (SD Fine Chemicals, Mumbai, India) was used as a solvent. As mobile phases for HPTLC analysis, toluene, acetone, and methanol (CDH Labs, Mumbai, India) were used. 2-amino ethyl diphenylborinate was provided by Sigma-Aldrich Co., LLC (St. Louis, MO, USA). All the solutions used in the analysis were filtered using a 0.22 µm syringe-driven filter (HIMEDIA, Mumbai, India). The following materials were obtained and used: Dulbecco’s modified eagle’s medium (DMEM), foetal bovine serum (FBS), MTT test kit, trypsin EDTA, Trypan blue solution, and 100 percent ethanol (Himedia Lab Pvt. Ltd., Mumbai, India). The HepG2 cell line was purchased from the National Centre for Cell Science (NCCS), an Autonomous Institute of the Department of Biotechnology, Government of India. Plant extracts were prepared using a Soxhlet extractor and hot percolation (Omega, Mumbai, India). The extracts were vacuum-dried on a rotary evaporator (BUCHI Rotavapor R-114, Houston, Texas, US). A CAMAG HPTLC system (Muttenz, Switzerland) equipped with a Linomat IV sample applicator was used for the HPTLC analysis of bioactive compounds. The extracts were applied to HPTLC plates (10 × 20 cm) pre-coated with silica gel 60F254 (Merck, Darmstadt, Germany). The HPTLC plates were developed in a CAMAG twin trough chamber (vertical development). Tissue culture flasks, 96- and 24-well micro-culture plates (Himedia Lab Pvt. Ltd., Mumbai, India), an Eppendorf tube, an inverted microscope, a serological pipette, a hemocytometer (Himedia Lab Pvt. Ltd., Mumbai, India), laminar flow hoods (Khera Instrument, New Delhi, India), and a CO2 incubator (NuAire, Fernbrook lane, Plymouth, USA) were used.

### 4.2. Collection of Plant Materials

Fresh plant material leaves were collected in March 2022 from the Angadippuram region of Malappuram district, Kerala, India. This practice was identified and authenticated by Dr.V.S. Hareesh, Research Officer at the KSCSTE-Malabar Botanical Garden and Institute for Plant Sciences, Kozhikode, Kerala, India. The specimen of the leaves was identified using the reference flora number (MBGIPS/09/2019-A1). The identified specimen was deposited at the herbarium of Malabar Botanical Garden and Institute for Plant Sciences (MBGIPS/09/2019-A1/2022) with accession number 7636.

### 4.3. HPTLC Instrumentation 

#### 4.3.1. Extraction of *Boerhavia diffusa*

The leaves were dried by air drying in the shade at room temperature (25 °C) and then ground into a coarse powder. Dried powdered plant material was subjected to extraction using alcohol and hydro-alcohol (50%) for 7 h at 37 °C. The combined alcoholic and hydro-alcoholic extracts were filtered separately and evaporated to dryness under reduced pressure at 40 °C with a rotary evaporator (BUCHI Rotavapor R-114, Switzerland). To obtain residues, extracts were placed into a china dish and maintained in a water bath at 40 °C. In total, 100 mg of crude extracts were accurately weighed, dissolved in 10 mL of HPLC grade methanol, sonicated for 10 min, and then made up with 10 mL of methanol. After filtering the solution, a 100 µL capacity Hamilton syringe was used to inject it into the HPTLC system.

#### 4.3.2. Preparation of Standard Solutions

The standard solution was developed by dissolving precisely weighed 1.0 mg of eupalitin-3-*O-β-D*-galactopyranosidein 1.0 mL of methanol HPLC grade as stock solution and storing it at 4 °C. These standards were further diluted to achieve the desired concentration for quantification.

#### 4.3.3. Preparation of the Plate

Prior to usage, pre-coated silica gel 60 F254 aluminium HPTLC plates (Merck, Darmstadt, Germany) were washed in methanol and dried. The standard solution of eupalitin-3-*O-β-D*-galactopyranoside and samples were applied on an HPTLC plate in the form of bands of 4 mm width using a Linomat V applicator (Sonnenmattstrasse, Muttenz, Switzerland) with a 100 µL syringe. The application rate was kept constant at 200 nL s^−1^, and the space between the two bands was 9 mm.

#### 4.3.4. Chromatography Parameters

The mobile phase was toluene: acetone: water (5:15:1“*v*/*v”*), and the development volumes were 20 mL. Linear ascending development was performed in a 20 × 10 cm twin-trough glass chamber saturated with the mobile phase. At room temperature, the optimal chamber saturation time for the mobile phase was 20 min. After development, the plates were dried in air and scanned at 340 nm using CAMAG HPTLC scanner 3 and Win CATS 4 software V 6.0 (Sonnenmattstrasse, Muttenz, Switzerland). The peak areas were recorded. 

#### 4.3.5. Calibration Curve of Eupalitin-3-*O**-β-D*-Galactopyranoside

A calibration curve with a standard concentration range of 100 to 5000 ng/spot was used to determine the content of the eupalitin-3-*O-β-D*-galactopyranosidecompound. In HPLC grade methanol, a stock solution of standard (1mg/mL) was prepared. Using an automatic sample spotter, different volumes of stock solution (5, 10, 15, 20, 25, and 30 µL) were spotted on an HPTLC plate to obtain concentrations of 100, 200, 400, 800, 1000, 2000, 3000, 4000, and 5000 ng/band. The area of each concentration peak was plotted against the amount of spotted or injected eupalitin-3-*O-β-D*-galactopyranoside. *R*^2^ = 0.9984 was determined for the linear regression of the standard curve. Y = 942.204 + 2.436X was the linear regression line. The regression data revealed a good linear relationship over the concentration range of 100–5000 ng/spot. The high correlation coefficient validates the linearity of calibration graphs and the system’s adherence to Beer’s law.

#### 4.3.6. Analytical Method Evaluation

The calibration curves’ lowest diluted solutions of the reference compounds were further diluted to a series of concentrations with HPLC grade methanol to determine limits of detection (LOD) and quantification (LOQ). Under the current chromatographic conditions, the LOD and LOQ were determined at signal-to-noise (S/N) ratios of 3 and 10, respectively. The standard calibration curves, regression equations, and LOD and LOQ values for each compound were validated under the ICH guidelines Q2 (R1) (2005). The previously analyzed samples were spiked with a standard at three different known concentration levels, namely 50, 100, and 150 percent, and the *B. diffusa* extract was re-analyzed using the proposed method to assess precision. The method’s precision was determined by performing intra-day and inter-day variation tests. The inter-day and intra-day variations for determining eupalitin-3-*O-β-D*-galactopyranosidewere carried out at different levels of concentrations of 100, 200, and 400 ng/band. The identification of bands was carried out in triplicate. The percentage R.S.D. was taken as a measure of precision. The specificity of the developed method for the analysis of eupalitin-3-*O-β-D*-galactopyranoside in the extracts was ascertained by the analysis of the standard and sample. The effects of small changes in mobile phase composition, volume, chamber saturation time, and solvent migration distance on the results were investigated. 

### 4.4. Optimization of Eupalitin-3-O-β-D-Galactopyranoside in Hydro-Alcoholic Extracts 

Three factors and levels were considered using the Box–Behnken statistical design. There were 17test trails. Minneapolis, Stat-Ease V6 software (Minneapolis, MN, USA) was used to optimize the design (Minneapolis, Stat-Ease Inc., MN, USA). This style is suitable for investigating and constructing second-order polynomial models and quadratic response surfaces. The designed style contains the replicated center point of the four-dimensional cube and a group of purposes lying at the center of every edge, confirming the region of interest. The dependent and independent variables are recorded in Table 4. The proposed HPTLC method was optimized to identify eupalitin-3-*O-β-D*-galactopyranosidefrom 17 Box–Behnken statistical design test trails.

The quadratic equation created by the Box–Behnken Design-Expert equation,
(1)R = A_0_ + A_1_ B_1_ + A_2_ B_2_ + A_3_ B_3_ + A_4_ B_1_ B_2_ + A_5_ B_1_ B_3_ + A_6_ B_2_ B_3_ + A_7_ B_1_^2^ + A_8_ B_2_^2^ + A_9_ B_3_^2^,
where A_0_ is the intercept, A_1_ to A_9_ are the regression coefficients, R is the dependent variable, and B_1_, B_2_, andB_3_ are the independent variables. The maximum percentage yield of eupalitin-3-*O*-*β*-*D*-galactopyranosidein the hydro-alcoholic extract of plant materials was recorded using the Design-Expert software system through arithmetic improvement. The Box–Behnken Design style spreadsheet is shown in Table 5.

### 4.5. Extraction and Isolation of Eupalitin-3-O-β-D-Galactopyranoside

Dried powdered plant material was subjected to extraction using hydro-alcohol (80% *v*/*v*) for 7 h at 37 °C. The combined hydro-alcoholic extracts were filtered and concentrated in a rotary evaporator under reduced pressure to obtain residues. Extracts were filtered and concentrated in a rotary evaporator under reduced pressure to obtain residues. The crude extracts obtained were fractionated with *n*-hexane (3 × 500 mL), chloroform (3× 500 mL), and ethyl acetate (3 × 500 mL). The dried ethyl acetate residues were thoroughly mixed in hot methanol and stored overnight in the refrigerator. The cold methanol was filtered, the solid was separated, and hot methanol was used many times to re-crystallize the solid in order to obtain pure eupalitin-3-*O-β-**D*-galactopyranoside. Yellow amorphous solids were characterized as eupalitin-3-*O-β-**D*-galactopyranoside using spectroscopic studies such as UV, NMR, and MASS and via comparison with published data. They were also characterized as eupalitin-3-*O*-*β*-*D*-galactopyranosidewhen matched with an authentic sample of eupalitin-3-*O-β-**D*-galactopyranoside by HPTLC.

### 4.6. Hepatoprotective Activity of Eupalitin-3-O-β-D-Galactopyranoside

In each well of the 24-well plates, cells were seeded at a density of 1 × 10^5^ cells/well and incubated overnight. After 24 h, the media was flicked off, and the cells were treated with eupalitin-3-*O-β-D*-galactopyranoside (ethyl acetate sediment), ethyl acetate supernatant, and chloroform fraction (100, 500, 1000 µg/mL) in separate wells of a 24-well plate and incubated for 2 h. Silymarin (100, 500 µg/mL) was used as a reference standard. After incubation, the cells were treated with a concentration of carbon tetrachloride (0.1%) and allowed to incubate for 2 h. Following incubation, the cells were washed and incubated for 1 h with MTT (20 μL of 5 mg/mL MTT in PBS) in each well. After 2 h, the formation of formazan crystals was observed under a microscope. If crystal formation was not successful, incubation was extended for another hour. The remaining formazan crystals were dissolved in 200 μL of DMSO in each well after removing the media. The cell culture plate was shaken for 15 min before the absorbance was measured with an ELISA reader at 540 nm. The percentage of hepatoprotection of eupalitin-3-*O-β-D*-galactopyranosideand silymarin was obtained using the following formula:
% Hepatoprotection = (Optical Density of Test sample/Optical Density of Control) × 100.

### 4.7. Statistical Analysis of Data

Graph Pad Software Inc., San Diego, CA, USA) was used for the statistical analysis. The results of animal trials were expressed as the arithmetic mean minus the standard error of the mean (S.E.M). One-way analysis of variance (ANOVA) was used to assess the effects of the various groups, and Dunnet’s multiple comparison tests were used to compare the effects of the treatment groups to the toxic control group.

## 5. Conclusions

The selected plants were collected in March 2022 and authenticated by Dr. V.S. Hareesh, Research Officer at the KSCSTE-Malabar Botanical Garden and Institute for Plant Sciences, Kozhikode, Kerala, India. The leaves of selected medicinal plants were first standardized as per the WHO guidelines. In the following stage, leaves were extracted using the Soxhlet extraction method with alcohol and hydro-alcohol (50% *v*/*v*) solvents for 6 h at 37 °C. The percentage yield was found to be the maximum in aqueous alcoholic extract followed by alcoholic extracts of selected medicinal plants, the probable reason being the presence of water-soluble polar compounds and solubility of polyphenols in aqueous alcoholic extracts. The prepared extracts were subjected to determination and quantification of eupalitin-3-*O*-*β*-*D*-galactopyranoside using the HPTLC method, which were successfully validated and developed. The method was validated in terms of linearity and detection limit, quantification limit, range, precision, specificity, accuracy, and robustness. The percentage yield of eupalitin-3-*O-β-D*-galactopyranoside was found to be the maximum in hydro-alcoholic extracts compared with alcoholic extracts, the probable reason being the presence of water-soluble polar compounds and the solubility of flavonoid glycosides in aqueous-alcoholic extracts. The percentage yield indicated that the selected medicines are a rich source of flavonoid glycosides. The flavonoid glycosides represent naturally occurring compounds with antioxidant and hepatoprotective potential. The best extraction method for the maximum percentage yield of eupalitin-3-*O-β-D*-galactopyranoside from *Boerhavia diffusa* requires appropriate extraction parameters such as extraction temperature, extraction time, organic solvents, and water content, which can be achieved using the Box–Behnken statistical design, which provides a time–temperature–solvent ratio. The Box–Behnken statistical design was used to determine the best optimization method, which was found to be extraction time (90 min), extraction temperature (45 °C), and solvent ratio (80% methanol in water *v*/*v*) for eupalitin-3-*O-β-**D*-galactopyranoside. The maximum percentage of eupalitin-3-*O-β-**D*-galactopyranoside in hydro-alcoholic extracts (80% *v*/*v*) was suggested using the Box–Behnken statistical design. The dried hydro-alcoholic extracts were then fractionated using n-hexane, chloroform, ethyl acetate, and water. The dried ethyl acetate fraction was thoroughly mixed in hot methanol and stored overnight in the refrigerator. The cold methanol was filtered, the solid was separated, and hot methanol was used many times to re-crystallize the solid in order to obtain pure eupalitin-3-*O*-*β*-*D*-galactopyranoside. Spectroscopic studies identified yellow amorphous solids as eupalitin-3-*O*-*β*-*D*-galactopyranoside. It was also characterized as eupalitin-3-*O-β-**D*-galactopyranosidewhen matched with an authentic sample of eupalitin-3-*O-β-**D*-galactopyranoside by HPTLC. Standard silymarin ranged from 80.2% at 100 µg/mLto 86.94% at 500 µg/mL in terms of significant high hepatoprotection (cell induced with carbon tetrachloride 0.1%), whereas isolated eupalitin-3-*O-β-**D*-galactopyranoside ranged from 62.62 percentage at 500 µg/mL to 70.23 percentage at 1000 µg/mL. The eupalitin-3-*O-β-**D*-galactopyranoside showed the most significant hepatoprotective activity compared with standard silymarin in HepG2 cells induced with carbon tetrachloride toxicity. The findings support that it is a source of structurally unique flavonoid compounds that may offer opportunities for developing novel semi-synthetic molecules for more recent indications.

## Figures and Tables

**Figure 1 molecules-27-06444-f001:**
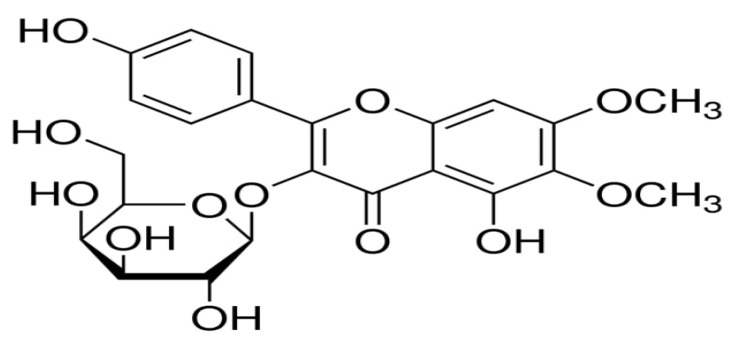
The chemical structure of eupalitin-3-*O**-β-D*-galactopyranoside.

**Figure 2 molecules-27-06444-f002:**
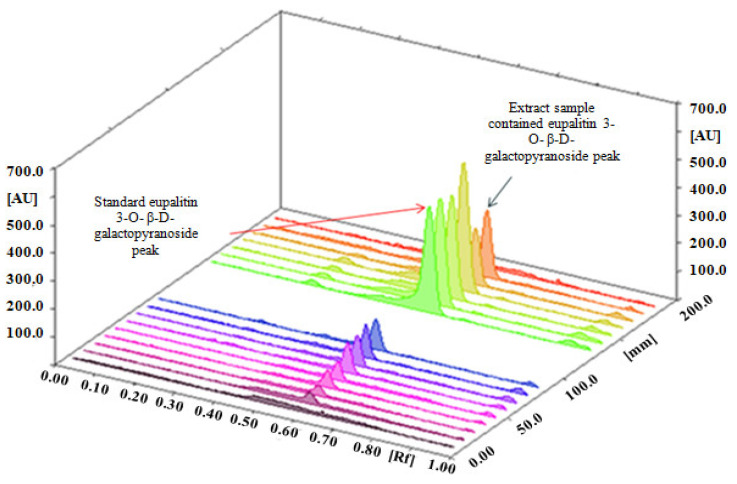
HPTLC densitogram of eupalitin-3-*O-β-D*-galactopyranoside and hydro-alcoholic extract in toluene: acetone: water (5:15:1) presented on HPTLC plate scanned at 366 nm.

**Figure 3 molecules-27-06444-f003:**
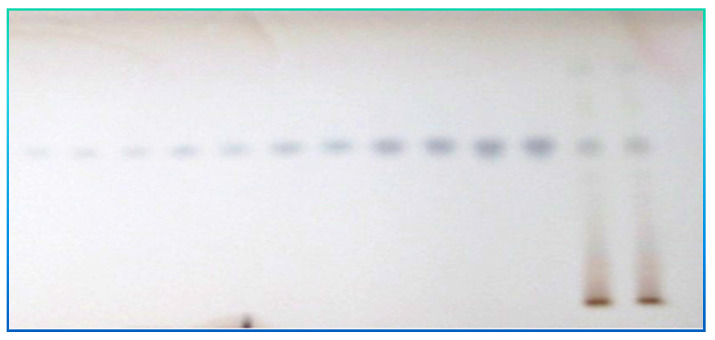
Chromatogram of eupalitin-3-*O-β-D*-galactopyranoside (R_f_ = 0.56).

**Figure 4 molecules-27-06444-f004:**
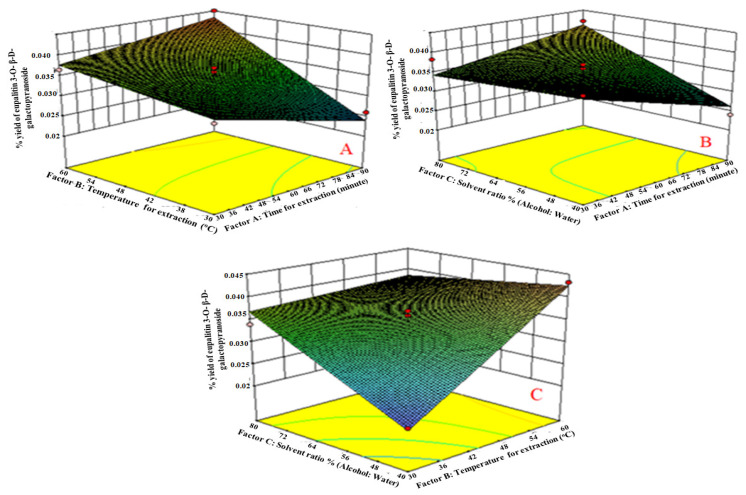
(**A**) Response surface plots of factor B_2_ vs. A_1_ against eupalitin-3-*O*-*β*-*D*-galactopyranoside: when time of extraction increases, the % yield of eupalitin-3-*O-β-D*-galactopyranoside also increases due to time required to penetrate solvent into plant materials; (**B**) response surface plots of factor C_3_ vs. B_2_ against eupalitin-3-*O*-*β*-*D*-galactopyranoside: As the temperature increases, the % yield of eupalitin-3-*O*-*β*-*D*-galactopyranoside also increases, respectively, due to thermo stable compounds; (**C**) response surface plots of factor C_3_ vs. A_1_ against eupalitin-*3-O-β-D*-galactopyranoside: the solvent ratio was higher, and the yield of eupalitin-3-*O-β-D*-galactopyranoside was higher. This could be attributed to the compound’s medium polarity.

**Figure 5 molecules-27-06444-f005:**
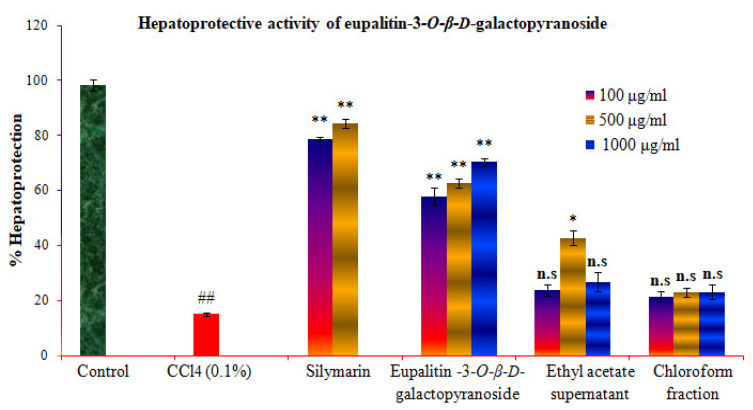
Effect of the eupalitin-3-*O*-*β*-*D*-galactopyranoside on CCl_4_-induced toxicity in HepG2 cells. HepG2 cells were incubated in the presence/absence of various compounds for 2h prior to treatment with CCl_4_ (0.1%) for 2h. Thereafter, the cells were processed for the MTT assay. The results are expressed as mean ± SEM. ## Carbon tetrachloride toxic group significant, “n.s” *p* > 0.05, ** *p* < 0.01, * *p* < 0.05 compared to the control group.

**Table 1 molecules-27-06444-t001:** Method validation parameters of eupalitin-3-*O-β-**D*-galactopyranoside.

Parameters	Eupalitin-3-*O-β-**D*-Galactopyranoside
LOD (ng)	30 ng
LOQ (ng)	100 ng
Specificity	Specific
Inter-day Precision (% RSD)	0.67
Intra-day precision(% RSD)	0.991
Regression equation	Y = 942.204 + 2.436 X
Linearity Range (Concentration)	100–5000 ng
Average recovery % RSD	99.78
R_f_ value	0.56
Correlation coefficient	0.9984

**Table 2 molecules-27-06444-t002:** Regression analysis results for models and responses (Y_1_) regression equations for the final suggested models.

Drugs	Models F Value	*R* ^2^	Adjusted *R*_2_	Predicted *R*_2_	SD	C.V.%
Eupalitin 3-*O-**β**-**D*-galactopyranoside (Y_1_)	Linear	0.540	0.4341	0.0799	0.00490	-
	2F_1_	0.890	0.7761	0.0181	0.00281	-
	Cubic	0.960	0.8231	-	0.00310	-
	Quadratic	0.8990	0.8200	0.5941	0.00281	8.04

**Table 3 molecules-27-06444-t003:** Effect of eupalitin-3-*O-β-**D*-galactopyranoside and standard silymarin on CCl_4_ induced hepatotoxicity.

Conc. (µg/mL)	Control	CCl_4_ (0.1%)	Standard Silymarin + CCl_4_	Eupalitin-3-*O*-*β*-*D*-Galactopyranoside + CCl_4_	Supernatant Ethyl Acetate + CCl_4_	Chloroform + CCl_4_
100	97.1 ± 2.2	15.2 ± 0.6 ^##^	80.12 ± 0.7 **	57.82 ± 3 **	23.84 ± 2.1 ^ns^	21.32 ± 1.9 ^ns^
500	-	-	86.94 ± 0.5 **	62.6 ± 1.5 **	42.7 ± 2.8 *	22.99 ± 1.6 ^ns^
1000	-	-	-	70.23 ± 1.5 **	26.66 ± 3.4 ^ns^	22.95 ± 2.6 ^ns^

ns: non-significant, *n* = 3, data±S.E.M. Groups I-4 were compared against group II using Dunnett’s post hoc test. ## Carbon tetrachloride toxic group significant, ns *p* > 0.05, ** *p* < 0.01, * *p* < 0.05.

**Table 4 molecules-27-06444-t004:** Independent and dependent variables selected in Box–Behnken design.

Factors Independent Variables	Levels Used		
	Low (−1)	Medium	High (+1)
A_1_ = Time in min	30	60	90
B_2_ = Temperature (°C)	30	45	60
C_3_ = Solvent ratio (% *v*/*v*)	40	60	80
Dependent Variables	**Goals**		
Y_1_ = Eupalitin-3-*O-**β**-**D*-galactopyranoside	Maximized		

**Table 5 molecules-27-06444-t005:** Mentioned responses in Box–Behnken design experiment for 17 analytical trails.

Run	Factor-1 (A_1_): Time (Min)	Factor-2 (B_2_): Temperature (°C)	Factor-3 (C_3_): Solvent Ratio (%)	% Yield Eupalitin-*3-O-**β**-**D*-Galactopyranoside (Y_1_)
01	30	30	60	0.0331
02	90	30	60	0.0234
03	30	60	60	0.0381
04	90	60	60	0.0429
05	30	45	40	0.0369
06	90	45	40	0.0259
07	30	45	80	0.0339
08	90	45	80	0.042
09	60	30	40	0.022
10	60	60	40	0.041
11	60	30	60	0.0369
12	60	60	80	0.0389
13	60	45	60	0.0351
14	60	45	60	0.0351
15	60	45	60	0.0339
16	60	45	60	0.0341
17	60	45	60	0.0359

## Data Availability

The data presented in this study are available upon reasonable request from the corresponding author.

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
