# Peer review of "A Box–Behnken Extraction Design and Hepatoprotective Effect of Isolated Eupalitin-3-O-β-D-Galactopyranoside from Boerhavia diffusa Linn."

_molecules, 2022, doi:10.3390/molecules27196444_

Round 1
Reviewer 1 Report
The authors presented a finding on extracting a bioactive flavonoid from Boerhavia diffusa. However, the manuscript is not well prepared, and several parts must be improved before the final decision.
1. There are a lot of formatting errors throughout the manuscript, including the title.
2. The title is hard to understand. Please revise
3. Eupalitin 3-O- β-D-galactopyranoside. The highlighted part should be italic, following the IUPAC format. Please correct it.
4. The study focuses on eupalitin 3-O- β-D-galactopyranoside; however, no chemical structure is provided in the manuscript. Please include the chemical structure in the introduction for ease of understanding.
5. Hard-to-understand sentences are throughout the manuscript. The manuscript needs to be proofread by a native speaker.
6. The introduction is not enough to justify the significance of this study. And why do the authors want to extract the targeted flavonoid only? What is the economic value of this compound? It is not well explained and justified!
Author Response
The authors would like to thank the esteemed reviewer for the criticism, valuable suggestions, and comments

Reviewer 2 Report
Although the manuscript has interesting subject, in my opinion, needs to be revised before it can be accepted for publication.
Abstract – every time abstract should contains the most important information like most important findings and results. Some values are needed. The abstract should be reorganized.
Introduction section - the Authors should try to make an effort to emphasize the importance of their studies. More details about the species used in the study are needed.
Materials and Methods – Collection of plant material - Please add the geographical coordinates of occurrence of the tested species. Which reference flora was used to identify the species?
It is not clear whether the samples were dried and in which conditions the drying took place. Please explain. How did the Authors follow the loss of water and the reproducibility?
More details on the HPLC analysis is needed.
Please revise the units throughout the manuscript.
The conclusions should be integrated with more detailed results summarizing all the study and must reflect the innovation of this study and the perspectives.
English and style require a careful reorganization. There are a lot of language mistakes, grammatically and stylistically.
Author Response

(The authors gave the same response as above.)

Reviewer 3 Report
Authors described the extraction parameters to recovery high yield of eupalitin 3-O- β-D-galactopyranoside from Boerhavia diffusa Linn leaves. The study is interesting and it can be considered for the publication in Molecules with major revisions.
General comments: in all manuscript space missing between words, starting from title, keywords and so on (check carefully in all manuscript) and remove space where not necessary. Check the correct form to write the compound name: β by italic, and D- by small caps (check in all manuscript).
The first sentence of teh abstract (lines 17-20) should be checked, is too long an dthe name of the compound was repeated three times. Introduction is well organized and cited references reported the correct background for the study. Botanical authority of all plant name should be cited the first mention
The experimental design is not clear. All extracts were fractionated? the extraction method and alcohol used were not reported as well as the percentage of the alcohol in the hydroalcoholic mixture. The cells used are HepG2 (add in the Material and methods section); they are tumoral cells, but in this study they were used as model cells? What does the choice of concentrations depend on? Probably a MTT before the hepatoprotection could be appropriate.
Use the correct order of the Table in the manuscript; the first cited Table is Table 3, check the order, please. The figure is not well clear, check the resolution. there are a lot of inaccurancies: some of them are the following: Hexane in n-hexane; ml in mL; Lines 36-40 check the sentence; line 89 check the concentration of the working standard and so on.
Check the style of references, is not the same.
Author Response

(The authors gave the same response as above.)

Round 2
Reviewer 1 Report
The authors made a significant improvement to the manuscript. However, some minor errors can still be identified throughout the manuscript.
1. Title: Hepatoprotective should be in small capital letters.
2. Line 64-65: Please make the plant name is correct (spacing).
3. Figure 1 (spacing): please revise the chemical structure of eupalitin-3-O-β-d-galactopyranoside to show the stereochemistry of sugar molecules (to differentiate galactoside and glucoside).
4. Line 107: plant name should be in italics.
5. Line 113: in vitro should be in italics.
6. Line 163: correlation coefficient (R2) - R2 should be written as R2
7. Please include a TLC chromatogram of the extract with the standard eupalitin-3-O-β-d-galactopyranoside to show the Rf value and the detected eupalitin in the extract.
8. Line 286: in vitro should be in italics
9. Line 288: CTC50 should be written as CTC50
10. Line 270: Please provide the full NMR (1H and 13C) data in a summarised form, for example, 1H NMR (MeOD): xxx; 13C NMR (MeOD) xxx
11. Please ensure the spacing between the number and measurement unit: 500 mL. Not 500mL. 33 % but not 33%
Author Response
The authors would like to thank the esteemed reviewer for the criticism, valuable suggestions, and comments.

Reviewer 2 Report
The authors only partially corrected the manuscript.
Author Response

(The authors gave the same response as above.)

Reviewer 3 Report
The manuscript has been improved by authors and in my opinion, it can be considered for the publication after minor revisions.
The letter D of the compound is not lowercase and not uppercase, but in small caps.
line 64-65: check the name of Boerrhaviadiffusa and Mamordicsubangulata, Naragamiaalata;
line 86: final dot was missing
line 97 (Figure1): add the space between figure and number;
lines 133 and 286: in vitro by italics.
General comments: the manuscript need to be revised, there are some inaccurancies as missing space after dot, or between number and measure unit. check the abbreviated name of journal (with or without final dot) in the list of references.
Author Response

(The authors gave the same response as above.)
